# Study on the Low-Temperature Pre-Desulfurization of Crumb Rubber-Modified Asphalt

**DOI:** 10.3390/polym15102273

**Published:** 2023-05-11

**Authors:** Shibo Zhang, Yang Yang, Rongxin Guo, Yong Yan, Haiyang Huan, Bangwei Wan

**Affiliations:** 1Yunnan Key Laboratory of Disaster Reduction in Civil Engineering, Kunming 650500, China; 20212210036@stu.kust.edu.cn (S.Z.); guorx@kmust.edu.cn (R.G.); yanyong@stu.kust.edu.cn (Y.Y.); 20212210045@stu.kust.edu.cn (H.H.); 20212210041@stu.kust.edu.cn (B.W.); 2Faculty of Civil Engineering and Mechanics, Kunming University of Science and Technology, Kunming 650500, China

**Keywords:** low-temperature desulfurization, high-solubility rubber, desulfurization mechanism, preparation process, asphalt mixture properties

## Abstract

Waste tires can be ground as crumb rubber (CR) and incorporated into asphalt pavement for efficient resource utilization. However, due to its thermodynamic incompatibility with asphalt, CR cannot be uniformly dispersed in the asphalt mix. In order to address this issue, pretreating the CR with desulfurization is a common way to restore some of the properties of natural rubber. The main technique of desulfurization and degradation is dynamic desulfurization, requiring a high temperature that may lead to asphalt fires, aging, and the volatilization of light substances, generating toxic gases and resulting in environmental pollution. Therefore, a green and low-temperature controlled desulfurization technology is proposed in this study to exploit the maximum potential of CR desulfurization and obtain high-solubility “liquid waste rubber” (LWR) close to the ultimate regeneration level. In this work, LWR-modified asphalt (LRMA) with superior low-temperature performance and processability, stable storage, and less susceptibility to segregation was developed. Nevertheless, its rutting and deformation resistance deteriorated at high temperatures. The results showed that the proposed CR-desulfurization technology could produce LWR with 76.9% solubility at a low temperature of 160 °C, which is close to or even better than the finished products produced at the preparation temperature of TB technology, i.e., 220–280 °C.

## 1. Introduction

Incorporating crumb rubber (CR) into asphalt pavements for improved properties and efficient waste utilization has gained significant attention recently. The use of asphalt pavements modified by adding ground waste-tire CR to the asphalt mixtures alleviates the issues of low-temperature cracking and high-temperature rutting of roads [1,2,3]. Pavements with rubberized asphalt mixtures are high performance and have a long service life, exhibiting excellent resistance to low-temperature cracking and high-temperature rutting while improving the utilization and economic value of scrap-tire resources [4]. However, the performance of CR-modified asphalt is easily degraded due to the poor compatibility of CR and asphalt as well as its poor storage stability [5,6,7], limiting its application in pavements. Therefore, more and more studies are focusing on the CR-desulfurization process to improve the performance and stability of CR-modified asphalts [8,9].

Gum-powder desulfurization in a liquid environment has shown improved results [10]. Dong et al. [11] investigated the feasibility of using waste rubber/oil (WRO) produced by photothermolysis of shredded tire rubber (CTR) as a bitumen modifier using waste cooking oil as a desulfurization solvent. It was confirmed that the storage stability, low-temperature plasticity, and processability of WRO-modified asphalt were considerably improved compared to rubber asphalt. Therefore, in this study, a more consistent and inexpensive source of naphthenic oil (AO) was chosen as the CR-desulfurization solvent [12] to achieve better desulfurization results. Zhang et al. [13] adopted the rubber powder desulfurization approach and evaluated the physical properties of modified asphalt by considering the influence of desulfurization process variables such as the type and amount of desulfurizing agent and the temperature and duration of desulfurization. It was shown that mixing rubber powder with the desulfurizing agent could reduce the viscosity of CR-modified asphalt. Rasool [14] et al. used pre-desulfurized highly reclaimed rubber (HRR) to improve the aging resistance and physical properties of styrene butadiene styrene (SBS)-modified asphalt (HRRMA). Presti et al. [15] attempted to unlock the full potential of desulfurized tire-rubber-heavy oil blends by tailoring recycled polymer-modified asphalt with unconventionally high CR content. The results showed that the available temperature range of the base asphalt was significantly improved by maintaining the solubility values of the CR in the asphalt so that they were considered stable at the hot storage temperature. Rubber hydrocarbon content (solubility) is the dominant factor affecting the high-temperature elasticity of CR-modified asphalt; the higher the content, the more pronounced the elasticity of the modified asphalt [16]. Many studies have shown that the pre-desulfurization of CR enhances the properties of resulting asphalt mixtures and that increasing the solubility of desulfurized CR is conducive to achieving this goal. However, the high temperature required for CR desulfurization hampers its applicability.

This study aimed to produce LWR with high solubility at a lower temperature (160 °C) and explore the effects of lowering the desulfurization temperature of CR. The LWR prepared at different temperatures was characterized from macroscopic and microscopic perspectives, and the mechanism of desulfurization and performance differences was analyzed. LWR solubility tests, scanning electron microscopy (SEM), atomic force microscopy (AFM), and Fourier transform infrared spectroscopy (FTIR) were performed. Then, LWR prepared at different temperatures was added to the base asphalt to prepare a rubber–asphalt mixture (LRMA). Considering the influence of LWR content on the asphalt mixture, the softening point, 5 °C ductility, 25 °C penetration, and Brinell rotation viscosity of LRMA were determined. The results showed that the prepared LRMA had good compatibility and exhibited better low-temperature performance than SBS-modified asphalt. However, the high-temperature performance was adversely affected by a decrease in the softening point and a deterioration in rutting resistance at high temperatures. This phenomenon is due to the regeneration degree of LWR near the limit, and losing its proper physical properties, directly affecting the high-temperature performance of CR-modified asphalt. Nevertheless, LRMA still has a greater research value and engineering significance due to its excellent low-temperature resource-recycling performance while solving the main storage stability problem in CR-modified asphalt.

## 2. Materials and Methods

### 2.1. Raw Materials

The raw materials required for this study include scrap tire rubber powder and industrial naphthenic oil AO; homemade catalyst DX. 60 mesh CR obtained from Shaanxi Hongrui Rubber Products Co. Ltd. was used in the study. These crumb-rubber particles were made from waste tires after removing metal and textile pollutants. The properties of the tires are shown in Table 1. In addition, the industrial naphthenic oil AO produced by PetroChina Karamay Petrochemical Company was chosen; its properties are given in Table 2. A homemade catalyst, DX (a mixture of 2,2′-dibenzoylaminodiphenyl disulfide (DBD) and zinc oxide (ZnO)), was also used. In order to prepare the asphalt mixture, 70#A grade matrix asphalt produced by Yunnan Petrochemical (Table 3), meeting the performance requirements specified in the technical specification for highway asphalt pavement construction JTG F40-2011, was used.

### 2.2. Preparation of Liquid Waste Rubber (LWR)

In order to achieve a CR pre-desulfurization treatment with simple operation and low equipment requirements, a 4:6 ratio of gum powder to black oil in the sample was selected in this study. First, the pre-weighed gum powder was placed in an iron vessel. Then, 3% of catalyst DX (by total weight of the sample including the mass of CR and naphthenic oil AO) was weighed and added to the vessel. It was then thoroughly mixed with the gum powder. Next, the AO was heated in an oven until it changed to a liquid with good flowability, and then it was mixed with the gum powder in proportion. Finally, the LWR samples were made by placing the iron vessels containing the gum–oil mixture into a constant-temperature oil bath and stirring at 450 rpm for 2 h at different design temperatures (140–260 °C).

### 2.3. Preparation of LWR-Modified Asphalt (LRMA)

Different mixtures were prepared in this study based on the solubility data obtained from LWR extraction in a Soxhlet extractor for eight hours. LWR1 with DX addition at 160 °C, LWR2 without DX addition at 160 °C, and LWR3 without DX addition at 220 °C were chosen to characterize the effect of catalyst DX on CR desulfurization at different temperatures. Next, the LWR-modified asphalt (LRMA) was prepared by blending the selected LWR1, LWR2, and LWR3 into the matrix asphalt at different dosages (10%, 20%, 30%, and 40%) by means of shearing in an oil bath at 160 °C for 50 min at a rate of 4000 rpm and curing in an oven at 150 °C for 30 min. LRMA (1–12) was considered for the subsequent performance analysis. Figure 1 shows the schematic diagram of the preparation process of LWR and LRMA.

### 2.4. Solubility Test

The crosslinked network structure and vulcanization bonds of vulcanized rubber can be destroyed during the mechanochemical treatment. Thus, the molecular weight of vulcanized rubber is reduced, and linear rubber macromolecular chains soluble in organic solvents are produced. The higher the desulfurization degree of the rubber powder, the more thoroughly the crosslinked network structure is destroyed, and the more soluble molecules are produced. Therefore, the number and solubility of soluble molecules can reflect the regeneration degree and desulfurization effect of rubber [17]. In this study, the solubility test was carried out to characterize the desulfurization degree of LWR. The test apparatus was a Soxhlet extractor, and the trichloroethylene recorded in JT/T 797-2019 was selected as the extractant. Nylon mesh with a pore size of 50 um was used as the filter paper in this test. First, the nylon net mass m1 was weighed on an electronic balance, followed by the LWR sample wrapped and tied tightly with the nylon net, weighed as *m*_2_ and placed in the extraction tube. Then, 250 mL of trichloroethylene solution (used as an extractant) was poured into a round beaker, and the heating device was turned on. The tap was opened to enable the condenser cycle. The extraction rate was controlled as the extractant refluxed approximately 6 times per hour for 8 h. After extraction, the sample was removed, and the nylon net residue was heated for 2 h using an oven to dry the solvent completely; it was then weighed as *m*_3_. The solubility of the gum powder can be calculated from Equation (1).
(1)$$\varphi =\left\{1-\left(m_{3}-m_{1}\right)/\left[\left(m_{2}-m_{1}\right)\times \varepsilon \right]\right\}\times 100\%$$
where ε is the proportion of gum powder in the gum–oil mixture.

### 2.5. SEM

The scanning electron microscope (Hitachi Regulus 8100) was used to observe the micromorphology and spatial structure of different LWRs and characterize the mechanism of LWR low-temperature desulfurization and the active form of LWR-modified asphalt [18].

### 2.6. FTIR

The NicolaetiS10-model instrument was used for the infrared analysis of LWR. The scanning range of the infrared spectrum was 600–4000 cm^−1^, and the resolution was 4 cm^−1^. The changes in the main functional groups of rubber powder before and after desulfurization were analyzed. The spectral band areas of different corresponding peaks were compared to further reveal the mechanism of LWR cracking during low-temperature desulfurization.

### 2.7. LRMA Performance Test

#### 2.7.1. Physical Properties of Modified Asphalt

The softening point, 5 °C ductility, 25 °C penetration, and Brookfield rotational viscosity of rubber powder-modified asphalt were tested following the Chinese Highway Asphalt Pavement Construction Technical Specification (JTG F40-2011) to characterize the properties of LRMA.

#### 2.7.2. Atomic Force Microscopy (AFM)

Morphological examination by AFM imaging is a sophisticated and easy-to-comprehend visual interpretation technique for assessing CR asphalt compatibility [19,20]. A Bruker Dimension ICON-model instrument was used to scan and track LRMA. All AFM scans were performed using tapping mode in air and at room temperature, 20 °C. The parameters used are shown in Table 4. The particle size of desulfurized CR, the two-dimensional topography map of the distribution shape, and the three-dimensional height phase map were obtained to clarify the surface structure, roughness, and aging resistance of CR-modified asphalt [21].

#### 2.7.3. Storage Stability Test

The storage stability of polymer-modified asphalt can be tested by segregation test [22], which is the difference between upper and lower softening points. The test was performed following JTG F40-2011 standard specifications. The sample was heated, and an aluminum tube was vertically placed into the bracket. A 50 g sample was weighed and poured into the open aluminum tube, and the opening was clamped. It was put in the oven at 163 °C for 48 h. Afterwards, the aluminum tube was removed and put in the refrigerator for 4 h. The upper and lower parts of the aluminum tube were recorded in the softening point test. The storage stability is considered good if the difference between the two softening points is less than 2.5. 

## 3. Results and Discussion

### 3.1. Preparation and Solubility Characterization of LWR

Various methods are used for CR desulfurization, e.g., chemical and biological methods (such as pre-reaction treatment, oxidation, polymer coating, solution immersion, and microbial reaction) and physical methods (such as plasma treatment and microwave desulfurization) [23,24]. In order to optimize the desulfurization process and reduce energy consumption, the mechanochemical desulfurization method for CR pre-desulfurization was used in this study. Inspired by the wet desulfurization process, industrial naphthenic oil AO—a petroleum byproduct—was used as a softener [25]. Its physical and chemical properties are similar to asphalt. It is a viscous liquid at room temperature with stable properties. It can better supplement a large amount of volatilized light components in the process of asphalt-mixture preparation. The selected catalyst DX is a disulfide regeneration activator that can selectively crack the vulcanized crosslinking bonds inside the rubber [26], maintain the integrity of the main bonds to a greater extent, and produce a good desulfurization effect [4]. The optimal ratio and dosage of catalyst DX were found as 1:1.2 and 3%, respectively. The catalyst DX was added to the mixture during desulfurization, followed by simple heating and stirring. CR was uniformly heated in AO during the reaction and fully reacted with catalyst DX. Therefore, the desulfurization temperature of CR could be reduced.

In this study, the desulfurization temperature was divided into different temperature ranges, i.e., low (140–180 °C), medium (180–220 °C), and high (220–260 °C) temperature ranges. As shown in Figure 2, when CR desulfurization is carried out at a low temperature (140 °C) without catalyst DX, it is difficult to achieve the expected effect, as its solubility is 0%; even the mass of the extracted residue is greater than that of the added rubber powder. The rubber powder was not successfully desulfurized and retained its original quality. Additionally, some substances were insoluble in trichloroethylene in AO and remained on the nylon mesh after extraction, increasing the residue quantity. The test results showed that at the desulfurization temperatures of 140 °C and 160 °C, the samples showed a macroscopic viscoelastic state, i.e., the formation of agglomerated small particle solids, if no catalyst DX was used.Combining this with the solubility results, it can be ascertained that at <160 °C desulfurization temperature, it is difficult to achieve the desulfurization regeneration of gum powder without adding catalyst DX (while using only AO as a softener). Within 160–180 °C, the effect of catalyst DX on CR desulfurization is particularly prominent, achieving a degree of CR regeneration that is difficult to achieve under normal conditions at this temperature, and its solubility is close to that of LWR prepared in the high-temperature range. This shows that the catalyst DX can effectively lower the desulfurization temperature and reduce the vulcanized rubber to a sol component with fluidity and plasticity. As the desulfurization temperature of the rubber powder is significantly reduced, the dense white smoke is not produced during desulfurization, and the amount of irritating gas is extremely small, reducing environmental deterioration. Therefore, the CR-desulfurization temperature of 160 °C was determined in this study.

The effects of gum–oil ratio and desulfurization duration on CR solubility were further studied. Five rubber-to-oil ratios (3:7, 4:6, 5:5, 6:4, and 7:3) were designed to simplify the CR-desulfurization process. The test results show that when the rubber-to-oil ratio is too small, the CR content in the LWR sample is small, and it is difficult for it to play a practical role in asphalt modification. However, when the rubber-to-oil ratio is too large, the sample becomes sticky and swells during operation, making it difficult to stir. Thus, the ideal desulfurization effect cannot be achieved. Therefore, three rubber-to-oil ratios of 4:6, 5:5, and 6:4 were selected in this study for further investigation. Regarding desulfurization time, 0.5 h, 1 h, 1.5 h, and 2 h were selected for research. The results are shown in Figure 3. When catalyst DX was used, the rubber-to-oil ratio was 4:6, while the desulfurization time was 2 h, and LWR could achieve higher solubility at a low temperature of 160 °C.

#### 3.1.1. Macroscopic Characteristics of LWR with Different Solubility

At the temperature selected in the experiment, there are disparities in the macroscopic state of the samples with catalyst DX added and without catalyst DX added, but both show the same trend as the solubility of LWR. As shown in Figure 4, the samples appeared as viscoelastic solid particles at desulfurization temperatures below 160 °C, even without using catalyst DX. Insoluble rubber particles are among the primary reasons for the unsatisfactory application of CR-modified asphalt or other rubbers [27]. Results have shown that CR is not highly compatible with asphalt in this state, and it is difficult for it to be uniformly dispersed in asphalt, resulting in the lower storage stability and performance of CR-modified asphalt [28]. In addition, this LWR content in asphalt usually only reaches 10% and cannot be used as a substitute for asphalt, resulting in low resource utilization. When catalyst DX is added, even if the desulfurization temperature is 160 °C, LWR solubility increases nearly three times. The sample presents a viscous fluid state close to the state of LWR obtained in a high-temperature environment. It is observed that there are only a small amount of CR particles, and the distribution is uniform and stable. The extraction residue has only a small amount of insoluble substances such as ash and carbon black. This shows that adding catalyst DX and softener AO can realize the potential of releasing CR in a low-temperature environment, making CR almost completely desulfurized, thereby improving the performance of LRMA. Therefore, the modification of asphalt with a high content of low-temperature desulfurized rubber powder may be studied further with a view to the maximum content of LWR in the asphalt mixture reaching 30–40%.

#### 3.1.2. Microscopic Analysis of LWR with Different Solubility

##### Scanning Electron Microscopy (SEM)

The micromorphological features of LWR samples during the desulfurization process were studied using SEM [29]. As shown in Figure 5, the structure and roughness of LWR after the pre-desulfurization treatment changed significantly. The LWR2 sample with low solubility (Figure 5c,d) has irregular surface morphology and high roughness, and there are continuous small, spherical particles and protrusions due to the CR particles and stirring paddles during desulfurization. Therefore, it can be ascertained that the desulfurization at this stage is primarily physical. By contrast, the LWR1 and LWR3 samples with higher solubility have smoother surface morphologies, lower roughness, greatly reduced spherical protrusions, and an evident “groove” structure. This is because the network structure of CR is rapidly destroyed during desulfurization, and the crosslinking bonds are broken, which leads to the shedding of many linear macromolecular chains released as sol components. These changes increase the contact area between LWR and asphalt, making it easier to absorb light components in asphalt and AO, thereby reducing the disparity in their densities. In addition, the particle size of the rubber particles is reduced during desulfurization, making the settling of small particles unlikely, thus improving the storage stability of the asphalt mixture.

It can be seen from Figure 5a,b,e,f that the microscopic morphologies of LWR1 and LWR3 are almost identical, and their solubility data are also very close, showing almost complete desulfurization of the CR. Thus, it is inferred that 160 °C temperature is sufficient for CR desulfurization when using catalyst DX, considering the preparation process of this study. Therefore, it can be suggested that the temperature increase slightly improves the degree of desulfurization of the CR.

##### Fourier Transform Infrared Spectroscopy (FTIR)

In order to further explore the desulfurization mechanism of LWR, FTIR was carried out on raw material (60 mesh CR) and LWR with varying solubilities. The corresponding FTIR spectra are shown in Figure 6. The position and peak position of the characteristic peaks of LWR with different solubilities do not change significantly, indicating that the breaking and formation of chemical bonds are relatively consistent. This shows that reducing the temperature does not negatively influence desulfurization. However, the functional group information of LWR is quite different from that of the raw-material CR, suggesting that the chemical structure of LWR has changed significantly compared with that of the raw-material CR. The disappearance of the absorption peak at 1537 cm^−1^ indicates that the aromatic ring C=C and N-H groups are destroyed. New antisymmetric stretching vibration absorption peaks appeared at 801 cm^−1^ and 1594 cm^−1^, representing aromatic NO_2_, possibly related to the added softener AO. Additionally, the diminishing double peaks at 1395 cm^−1^ and 1365 cm^−1^ are indicative of the destruction of the C(CH_3_)_3_ group, while the intensity of the absorption peak representing stretching vibration of the olefinic and conjugated olefinic double bonds gradually increased at 1600 cm^−1^. It confirms that the number of sol molecules in LWR increased.

### 3.2. LRMA Physical Properties

#### 3.2.1. Penetration, Softening Point, and Ductility

Brookfield rotational viscosity and softening point are more reasonable characterization methods for determining the high-temperature performance index of CR-modified asphalt, whereas the low-temperature performance is characterized by the 5 °C ductility test [30]. Figure 7 shows the performance indexes of LRMA and pure AO modification. The performance indexes of LRMA1 and LRMA3 with different LWR content are similar and superior to that of LRMA2. Penetration is an important indicator of asphalt softness and hardness. With increased LWR content, the penetration of three LRMAs shows a gradually increasing trend. In addition, the increasing penetration rates of LRMA1 and LRMA3 are greater than that of LRMA2. At 40% LWR content, the penetration of 0.1 mm of the former two has exceeded 20 mm, which may be attributed to the change in physical properties of CR due to an improved CR-desulfurization degree and the softening of the asphalt mixture due to AO addition [31]. The softening point and 5 °C ductility of asphalt correspond to the penetration. With the increased LWR content, the softening point of LRMA decreases, and the ductility increases greatly. The performance trends of the two are roughly in line with the linear law. When LWR content is 20%, the ductility of LRMA1 reaches the requirements of SBS-modified asphalt. When the content is 30% and 40%, the ductility reaches 32.9 cm and 65.9 cm, respectively, which is better than LRMA2 and LRMA3, showing excellent low-temperature performance. It can be concluded that when the LWR solubility reaches a similar level, the effect of desulfurization temperature on the recycled rubber is no longer significant. The LWR made at low temperatures still has a positive modification effect on asphalt, and there is no decline in the modification effect of LWR on asphalt due to the reduced desulfurization temperature. With the increased LWR content, the high-temperature performance of LRMA slightly decreased, but the low-temperature performance showed a significantly enhancing trend. Therefore, controlling the amount of LWR doping between 20% and 30% is recommended to achieve better performance.

#### 3.2.2. Ease of Fabrication

The viscosity of a rubber–asphalt mixture is directly related to the difficulty of its construction, reflecting to a certain extent the road performance of the asphalt mixture [32,33,34]. In this study, the variation model of Brookfield rotational viscosity of LRMA within 135–175 °C was observed. Figure 8 shows that the viscosity–temperature curves of different LRMAs follow the linear trend. LRMA viscosity decreases with the increase in LWR content. LRMA1 and LRMA3 have lower viscosities, showing similar curves, and the downward trend is gentle. This may be attributed to the high solubility of LWR1 and LWR3 and the presence of large amounts of sol components and disrupted vulcanization crosslinking networks, as well as the small amount of inorganic matter in LWR.

The decrease in LRMA1 and LRMA3 viscosities is attributed to the decreased particle size of the residual rubber powder in LWR and the uniform dispersion of the particles. LRMA2 is the typical desulfurized CR-modified bitumen produced without catalyst DX during desulfurization. Its viscosity is slightly larger, and the declining trend is steep, possibly because of the low LWR2 solubility. Thus, many CR particles in LRMA2 could not be desulfurized. These rubber powder particles can easily settle to the bottom of the asphalt, increasing the rotor resistance and causing uneven heating during the test. Consequently, the viscosity changes inconsistently. Comparing the results of LRMA1 and LRMA3, it is observed that LRMA1 obtained in a low-temperature environment of 160 °C can meet the workability requirements.

The viscosity–temperature error can also explain the CR dispersion in the asphalt mixture [17]. The viscosity error fluctuation range of LRMA2 is the highest, which is 464–720 mPa·s, whereas those of LRMA1 and LRMA3 are 226–271 mPa·s and 235–249 mPa·s, respectively. It can be seen that the LWR subjected to desulfurization treatment in different high- and low-temperature environments has good dispersion, indicating that highly soluble CR can be prepared in a low-temperature environment. 

#### 3.2.3. Storage Stability

Figure 9 shows the differences in softening points of LRMAs. There is a significant disparity in the segregation difference (the difference between upper and lower softening points) of LRMA. LRMA2 has been observed to have the highest segregation difference. When the LWR content is 10%, the segregation difference reaches 4.4 °C, which exceeds 8 °C when the LWR content is 20% to 40%. The segregation difference gradually increases with LWR dosage, which does not meet the requirement of less than 2.5 °C in the JTG F40-2011 specification. This indicates that there is a density difference between CR and virgin bitumen. Heavy CR particles have poor compatibility with lighter asphalt [35]. When dispersed in an asphalt mixture, they tend to settle to the bottom due to gravitational force; therefore, CR without desulfurization treatment directly incorporated for asphalt modification will have compatibility issues affecting the performance and storage stability of the asphalt mixture [36].

The segregation differences for both LRMA1 and LRMA3 were within specification. For the LWR, content is 10–40%; the interval differences are 0–1.9 °C and 0.2–2.3 °C, respectively. In particular, when the LWR content is 10–20%, the segregation difference is less than 1. This shows that LWR does not segregate in asphalt, which may be attributed to the similar density and state of LWR and asphalt. Additionally, the surface roughness of CR after desulfurization increases, and it is easy for it to have a new chemical reaction with asphalt. Its particle size and quality are significantly reduced, making sedimentation unlikely. In addition, the linear molecular chains and rubber shins in the sol part are also easily soluble in asphalt. Therefore, the high-solubility LRMA prepared in a low-temperature environment still has good storage stability.

#### 3.2.4. Anti-Aging Properties

The anti-aging properties of pre-desulfurized LWR-modified asphalt (LRMA) were further investigated using atomic force microscopy (AFM) [37,38,39]. Aging adversely affects the performance of asphalt mixtures and is characterized by surface roughness and a resulting honeycomb structure [40]. The results are shown in Figure 10. Through 2D surface scanning results and 3D morphology analysis, it has been found that LRMA2 contains a greater amount of large CR particles and has more chaotic bee-like structures, whereas LRMA1 and LRMA3 have fewer bee-like structures and are more gentle. Gwyddion software was used to calculate the average surface roughness of LRMA. Table 5 shows that the average surface roughness of LRMA2 is 0.649 nm, which is greater than that of LRMA1 and LRMA3. Therefore, it is deduced that LRMA with pre-desulfurization treatment has a better anti-aging ability. This may be because the catalyst DX added during CR desulfurization positively affects the anti-aging ability of the asphalt mixture. At the same time, during the high-speed shearing, the C=C double bonds in CR are largely opened and react with asphalt to form a new stable crosslinked structure, thereby improving the aging resistance of rubber powder-modified asphalt. Therefore, reducing the desulfurization temperature will not affect the anti-aging ability of LRMA, and the high solubility of LWR has a positive modification effect on the anti-aging performance of the asphalt mixture.

## 4. Conclusions

In this study, the mechanochemical method was used to lower the CR-desulfurization temperature and enhance the solubility of desulfurized CR by considering the influence of variables such as desulfurization method, catalyst type, content, and preparation process. The properties of CR-modified asphalt were tested after preparing a high-solubility rubber through low-temperature desulfurization. The conclusions drawn from the obtained results are as follows.

(1) Utilizing catalyst DX and softener AO for desulfurization can decrease the temperature required for rubber desulfurization and facilitate the modification of the crosslinked network structure of CR, ultimately leading to an improved degree of reduction and regeneration. At the preparation temperature of 160 °C, the solubility of CR can reach 76.9%, but the high solubility affects its physical properties.

(2) CR desulfurization is significantly influenced by the preparation process. During low-temperature desulfurization, the catalyst must interact with CR, and ample time for chemical reaction should be available. Therefore, a stirring rate of 450 rpm and a desulfurization duration of 1.5–2 h are recommended.

(3) The performance of LWR-modified asphalt prepared at high and low temperatures is similar, confirming that lowering the desulfurizing temperature does not affect the effectiveness of LWR in asphalt mixture, and the increased solubility positively impacts the asphalt mixture’s performance. The low-temperature performance and storage stability of desulfurized CR-modified asphalt are improved, but the high-temperature performance is significantly reduced. The amount of LWR content in the asphalt mixture also plays a critical role, and a 20–30% dosage is recommended.

## Figures and Tables

**Figure 1 polymers-15-02273-f001:**
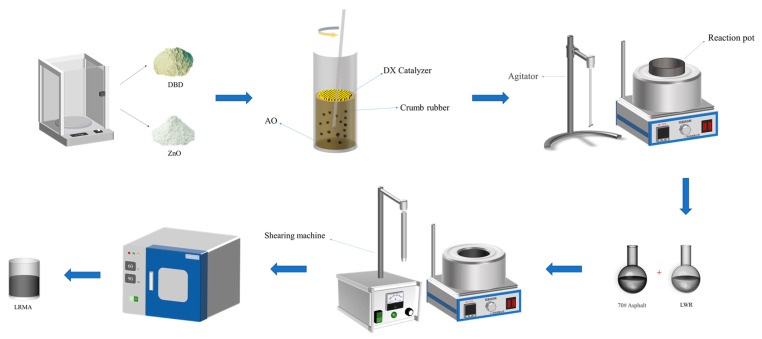
Schematic diagram of the preparation process of LWR and LRMA.

**Figure 2 polymers-15-02273-f002:**
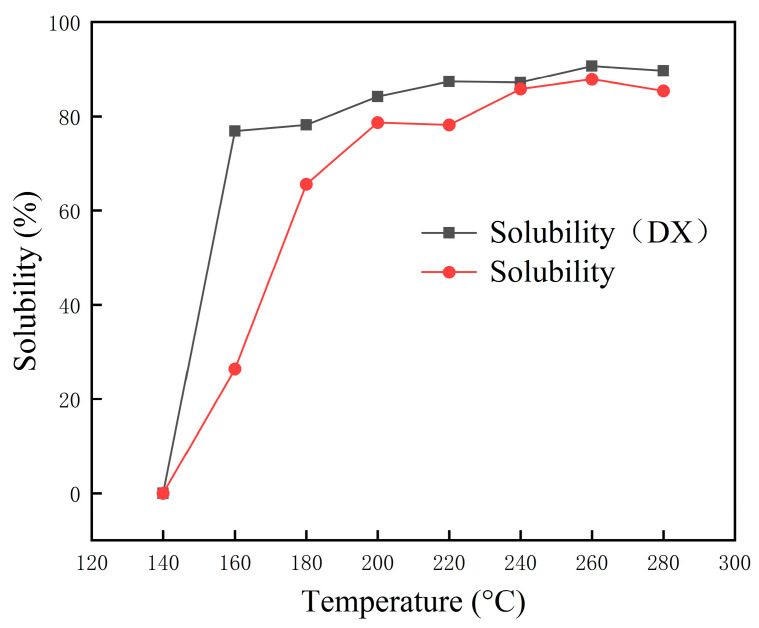
Solubility versus temperature for waste rubber powder.

**Figure 3 polymers-15-02273-f003:**
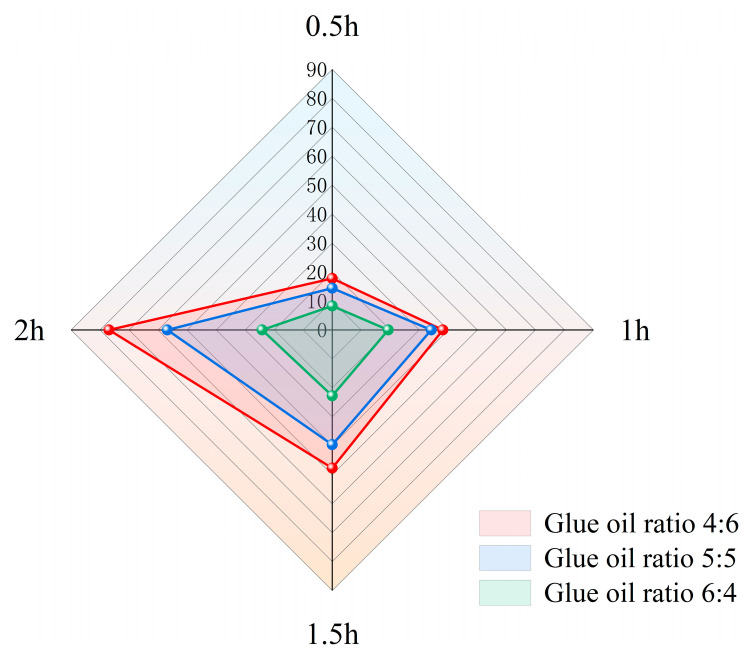
Solubility of LWR at different rubber–oil ratios and desulfurization times.

**Figure 4 polymers-15-02273-f004:**
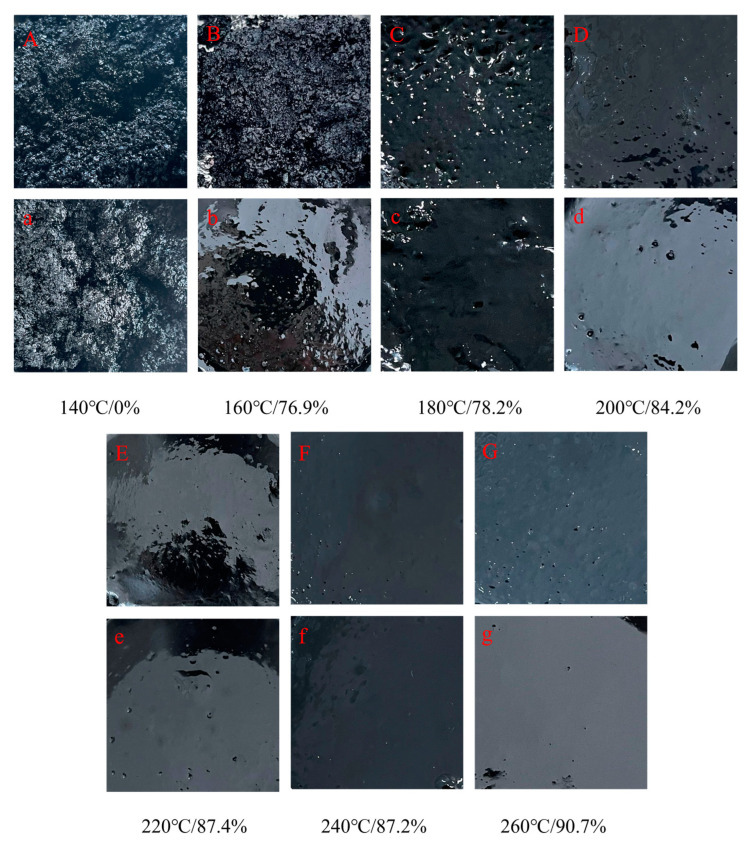
(**A**–**G**) is the apparent morphology of LWR without catalyst DX; (**a**–**g**) is the apparent morphology of LWR with catalyst DX.

**Figure 5 polymers-15-02273-f005:**
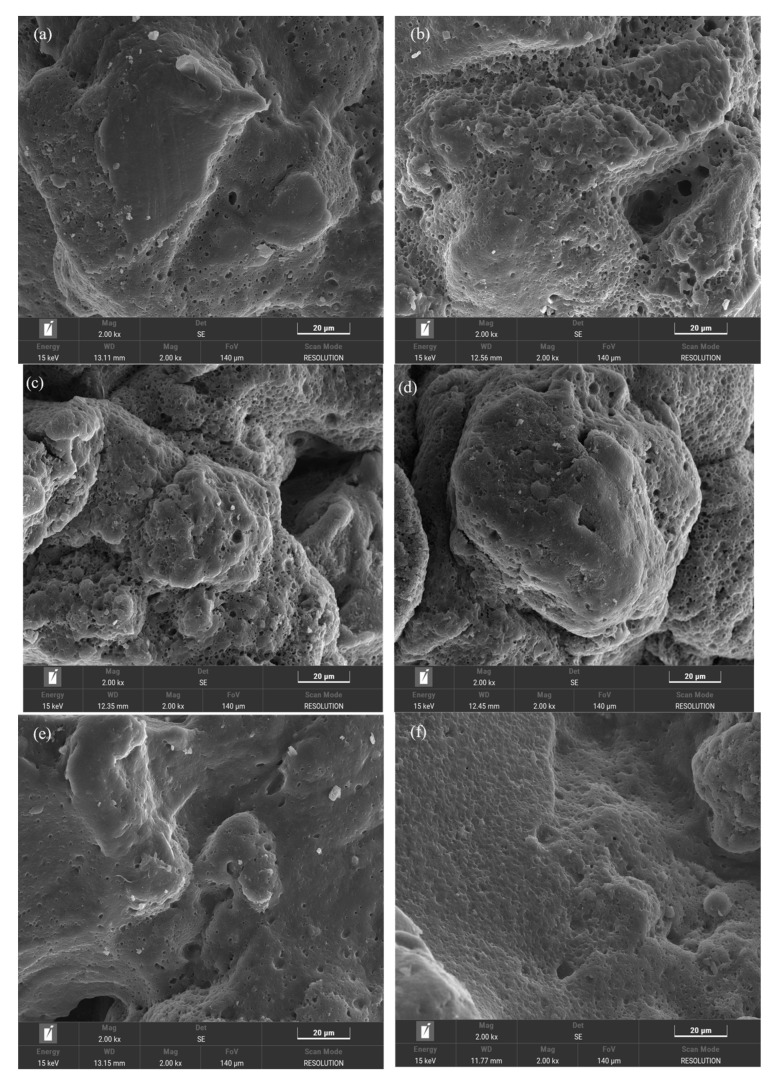
SEM: (**a**,**b**) LWR1; (**c**,**d**) LWR2; (**e**,**f**) LWR3.

**Figure 6 polymers-15-02273-f006:**
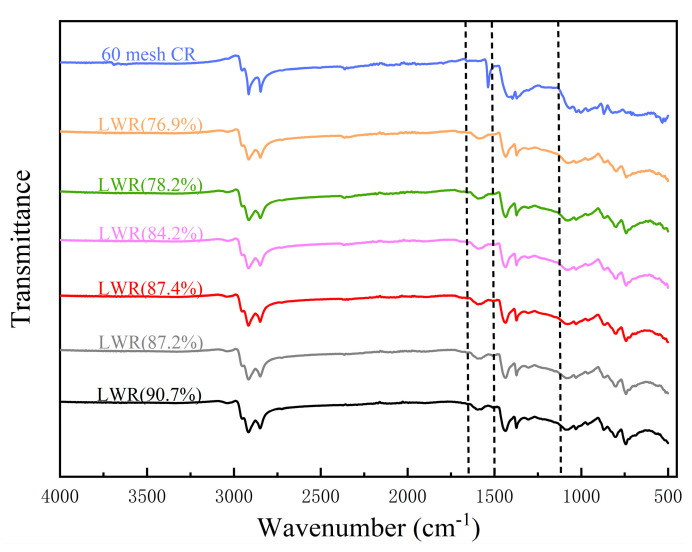
FTIR spectra of raw material (60 mesh CR) and LWR with varying solubility.

**Figure 7 polymers-15-02273-f007:**
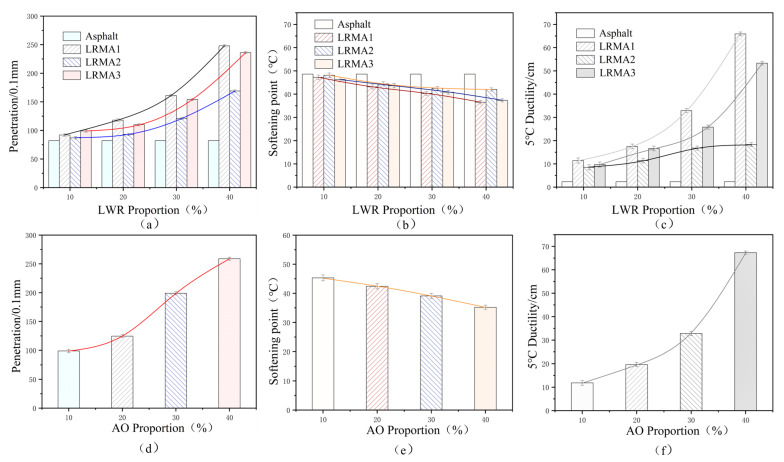
Physical properties of asphalt mixture and base asphalt: (**a**) permeability of LRMA, (**b**) softening point of LRMA, (**c**) 5 °C ductility of LRMA, (**d**) permeability of AO modified asphalt, (**e**) softening point of AO modified asphalt, (**f**) 5 °C ductility of AO modified asphalt.

**Figure 8 polymers-15-02273-f008:**
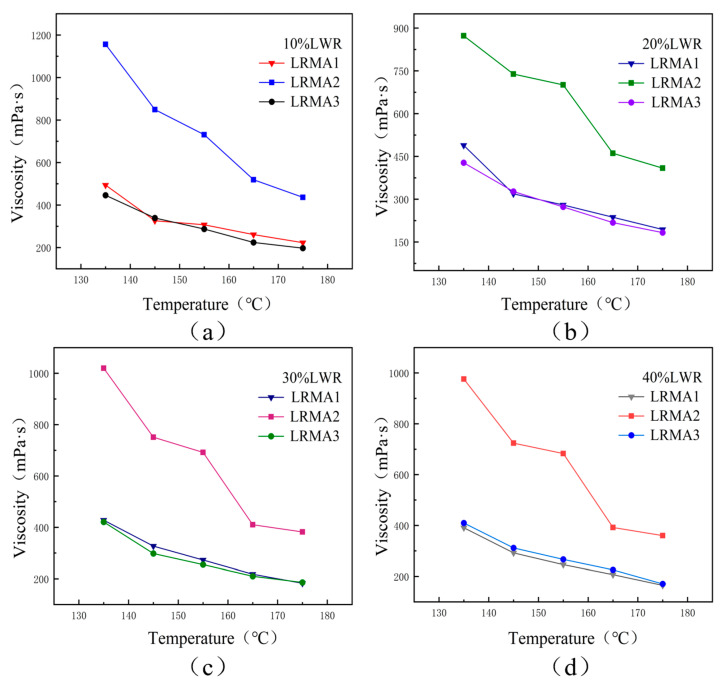
Viscosity-temperature curves of LRMA with different LWR doping (**a**) LRMA with 10% LWR, (**b**) LRMA with 20% LWR, (**c**) LRMA with 30% LWR, (**d**) LRMA with 40% LWR.

**Figure 9 polymers-15-02273-f009:**
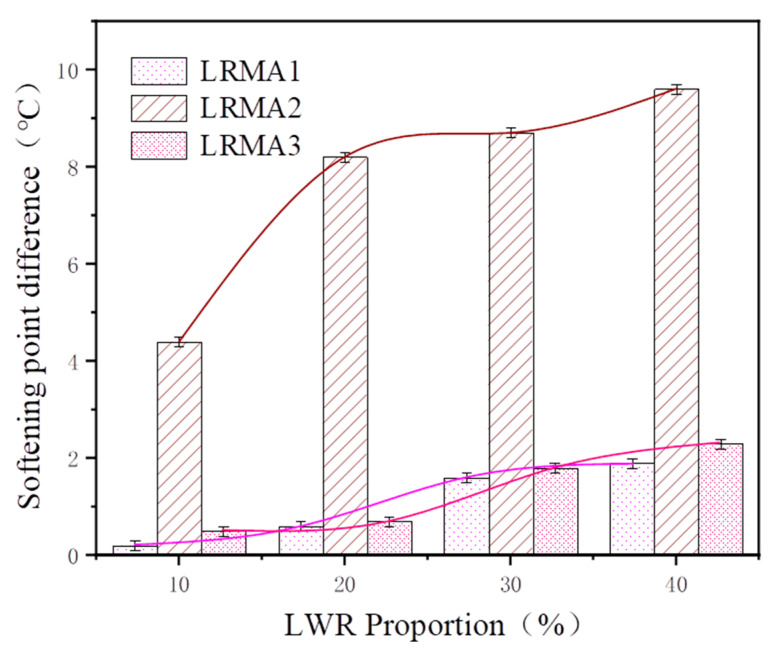
Differences in softening point of LRMA.

**Figure 10 polymers-15-02273-f010:**
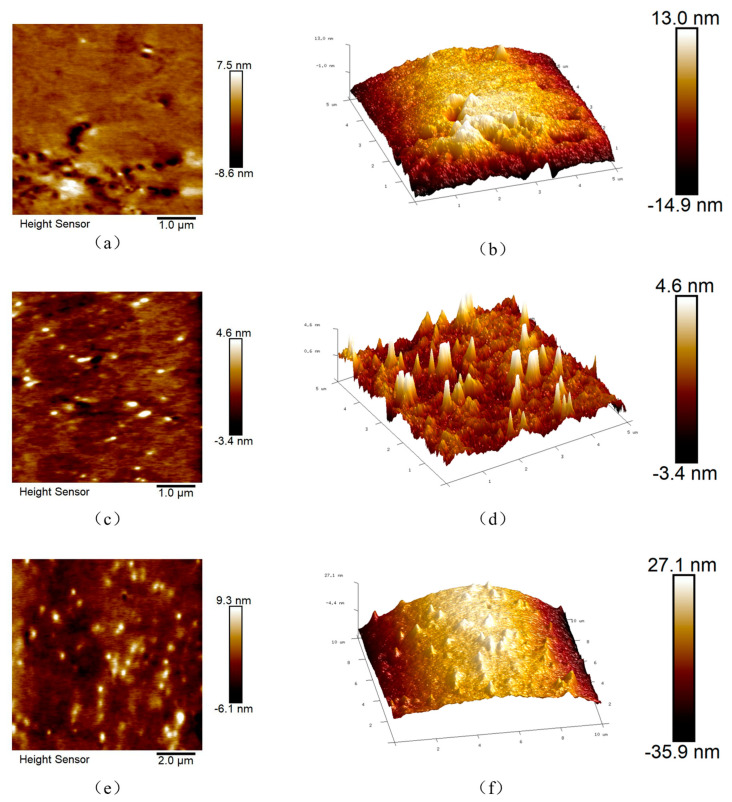
3D topography of LRMA1, LRMA2, and LRMA3-AFM: (**a**,**c**,**e**), and 2D appearance of LRMA1, LRMA2, LRMA3: (**b**,**d**,**f**).

**Table 1 polymers-15-02273-t001:** CR properties.

Index	Heating Loss (%)	Ash Content (%)	Iron Content (%)	Fiber Content (%)	Sieve Residue (%)	Density (g/cm^−3^)
**Test value**	0.62	7.75	0.029	0	0.014	1.19
**Standard value**	-	≤8	-	<1	-	1.10–1.30

**Table 2 polymers-15-02273-t002:** Basic properties of naphthenic oil AO.

Index	Density (g/cm^−3^)	Kinematic Viscosity (m^2^/s^−1^ 100 °C)	Effective Content (%)
**Test value**	1.02	40	87
**Specification**	GB/T1884-92	GB/T265-88	ASM-IP-2

**Table 3 polymers-15-02273-t003:** Basic properties of asphalt.

Index	Penetration (25 °C, 5 s,100 g)	Softening Point (°C)	10° Ductility (cm)
**Test value**	76	48.8	17
**Standard value**	60–80	≥45	≥15

**Table 4 polymers-15-02273-t004:** General settings of the AFM.

Scan size	5	μm
Scan rate	2	KHz
Scan angle	0	°
Scan speed	37.21	μm/s
Set point	720.00	mV
Drive amplitude	9.77	mV
Drive frequency	300	Hz

**Table 5 polymers-15-02273-t005:** LRMA surface roughness.

Sample	RA (Sa) [nm]
LRMA1	0.649
LRMA2	3.59
LRMA3	1.02

## Data Availability

The data presented in this study are available from the corresponding author upon reasonable request.

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
