# Peer review of "Study on the Low-Temperature Pre-Desulfurization of Crumb Rubber-Modified Asphalt"

_polymers, 2023, doi:10.3390/polym15102273_

Round 1
Reviewer 1 Report
In this contribution, the authors prepared desulfurized crumb rubber using catalysis and naphthenic oil at low temperatures. The resulting liquid waste rubber modified asphalt (LRMA) shows good storage stability and ductility. This topic is inspiring to the readership of Polymers. However, this manuscript overlooked several critical discussions. The following questions and comments should be addressed before making a further decision.
1. Although the title of this manuscript includes reaction mechanism, this research does not present insightful investigations on mechanisms of either desulfurization of crumb rubber or the interaction of desulfurized rubber and asphalt, i.e., dissociation of sulfur bridge or interaction between rubber and asphalt. Therefore, it is not necessary to emphasize the reaction mechanism in the title.
2. Following the first comment, the title includes the pre-desulfurized crumb rubber but does not add the naphthenic oil as the modifier. What is the contribution of naphthenic oil to the properties of modified asphalt, e.g., increasing the penetration and ductility, and decreasing the softening point? Can the author add the neat naphthenic oil modified asphalt to Figure 7 for comparison?
3. What is the melting point of naphthenic oil AO? In Section 2.2, the AO was heated until it changed to a liquid state. But in Section 3.1, the AO is a viscous liquid at room temperature.
4. In Section 3.1, the mass of the extracted residue is greater than that of rubber powders, and the increasing mass is attributed to the insoluble substances in trichloroethylene. What are the insoluble substances, and why can those substances reflux onto and remain in the extracted residue? Alternatively, is it possible the increasing weight is due to the residual trichloroethylene in the swollen crumb rubber?
5. How are the LRMA aged in Section 3.2.4, e.g., PAV, QUV, RTFO, extended heating, etc.? What are the morphologies and roughness of LRMA before aging?
6. Other amendments:
In Section 3.1, what is the ratio of 1:1.2 in the 14th line, DBD:ZnO?
The crumb rubber desulfurized at 140 °C is not depicted in Figure 4, though the discussion in Section 3.1.1 mentions "As shown in Figure 4, the samples appeared as viscoelastic solid particles at desulfurization temperatures below 160 °C."
In the discussion of FTIR, what is the source of N=H (or N-H?) in the 10th line in the crumb rubber?
Table 2, 100 °C
In Section 2.4, solvent instead of water in the 17th line.

Reviewer 2 Report
The manuscript focuses on and investigates the effect of desulfurization of crumb rubber in bitumen with a novel naphthenic oil at a relatively lower temperature than the usual process. It studies the storage stability and low-temperature properties as well as the high-temperature performance. In general, the methodology followed is adequate and the results are well explained. The text flow is appropriate and in general, the manuscript could be benefited from the following recommendations and clarifications provided below.
Page 2, ‘Rubber hydrocarbon…asphalt’. Please elaborate on this argumentation about solubility. How this can be linked with the chemical potential of each component and how can thermodynamically explain the higher elasticity?
Section 2.3. what are the detailed solubility data that were obtained and why were they later expressed in % instead of common solubility units?
Section 2.5. & 2.7.2. Please provide more details about the magnification and settings of the microscopies. For example, to bring it to an acceptable level you can consult other descriptions see i.e. https://doi.org/10.1016/j.micron.2021.103149
Section 3.1. How exactly the optimal ratio and dosage of the catalyst DX were chosen? Please explain.
Figure 2. The figure mentioned on page 5 (last row) and the given figure number are not in line. Please carefully check the whole manuscript and the text when referring to the correct figure and the given numbers for all the figures and modify them accordingly. Moreover, what the y-axis in the current Figure 2 represents and why is given in % in the next figure and not in common solubility units?
Table 4. How many replicates/images were utilised to extract i.e. the surface roughness and how is compared in terms of average values of different images used and their statistics i.e. standard deviation? This is rather crucial since the conclusions may be biased. For example, recent studies have utilised at least 9 images for extraction of quantitative microscopic evaluation which might be worth looking at see https://doi.org/10.1016/j.micron.2022.103294
Round 2
Reviewer 1 Report
I appreciate the authors' efforts. All my questions have been addressed.
Minor English editing is recommended.
For example, "Also, some substances were insoluble in trichloroethylene in AO, which remained on the nylon mesh after extraction, increasing the residue quantity. (Section 3.1)" This description is unclear, leading to confusion in Comment #4.
Reviewer 2 Report
Thank you for the revision.